# Prolapsed Atypical Polypoid Adenomyoma—A Case Report and Literature Review

**DOI:** 10.3390/life13122352

**Published:** 2023-12-15

**Authors:** Tudor Butureanu, Ana-Maria Haliciu, Ana-Maria Apetrei, Ioana Pavaleanu, Razvan Vladimir Socolov, Raluca Anca Balan

**Affiliations:** 1Department of Mother and Child, “Gr. T. Popa” University of Medicine and Pharmacy, 700115 Iasi, Romania; tudorandreib@gmail.com (T.B.); ioana_pavaleanu@yahoo.com (I.P.); razvan.socolov@gmail.com (R.V.S.); 2“Elena Doamna” University Hospital of Obstetrics and Gynecology, 49, Elena Doamna St., 700398 Iasi, Romania; raluca.balan@umfiasi.ro; 3Department of Morpho-Functional Sciences I, “Gr. T. Popa” University of Medicine and Pharmacy, 700115 Iasi, Romania

**Keywords:** adenomyosis, polypoid, endometriosis, cervical, malignant, polyp, bleeding, vaginal

## Abstract

Atypical polypoid adenomyoma (APAM) is a rare polypoid benign tumor of the uterus that causes irregular vaginal bleeding in women of reproductive age. It has the potential for malignant transformation, but it does not metastasize. APAM may coexist with endometrial hyperplasia and adenocarcinoma, usually leading to misdiagnosis. Histopathologically, it is a biphasic tumor, represented by the endometrioid glands with a complex histoarchitecture, with sometimes squamous morular metaplasia or cytologic atypia, interspersed with a fibromyomatous stroma. This tumor has a high incidence of recurrence. We present a very rare case of a 21-year-old patient, a virgin, without a significant medical history, with a bleeding mass occupying the vagina. The mass was excised using forceps, scissors, and a suture of the visible pedicle. After a four-year follow-up and no additional medical treatment, no relapse was observed. Given the risk of recurrence and progression, APAM might be treated via a hysterectomy in patients with no desire for pregnancy. Due to a lower recurrence rate, the conservative treatment of atypical polypoid adenomyoma performed via an operative hysteroscopy represents the best choice. Previously diagnosed in hysterectomy specimens, with the introduction of better-performing indirect imaging techniques, adenomyosis is a clinical entity that has the possibility of being diagnosed in the presurgical stage.

## 1. Introduction

The 2020 World Health Organization (WHO) Classification for the mixed epithelial and mesenchymal tumors of the uterus encompasses various categories, including carcinosarcoma, adenosarcoma, adenofibroma, adenomyoma, and atypical polypoid adenomyoma. The latter two conditions, adenomyoma, and atypical polypoid adenomyoma, consist of a combination of benign epithelial and mesenchymal components with a significant presence of smooth muscle tissue [1,2].

Atypical polypoid adenomyoma or atypical polypoid adenomyofibroma of the uterus (APAM) is an uncommon uterine lesion that was first described by Mazur in 1981 [3].

APAM is a rare disease, with <500 cases reported in the literature, and therefore, it is difficult to evaluate its incidence. 

Rare examples of APAM have been described in patients with Turner syndrome who have been prescribed unopposed estrogens [3].

This condition is most frequently observed in fertile nulliparous women, typically occurring at an age between 21 and 73 years, with a median age of 38 years. Additionally, it is often correlated with infertility [4]. Conservative management is often chosen to preserve fertility in cases of atypical polypoid adenomyoma. However, it is important to note that there is a substantial risk of persistence or recurrence of this condition, which can be as high as 59% after a 10-year follow-up period. This risk of recurrence decreases to 22% when the condition is treated with a hysteroscopic-guided curettage, as opposed to a ‘blind’ dilation, a curettage, and a polypectomy [4].

It is worth emphasizing that patients with atypical polypoid adenomyoma face a higher overall risk of developing atypical hyperplasia and endometrioid adenocarcinoma of the endometrium compared to those with endometrial polyps. This underscores the importance of careful clinical management in this specific patient population [5]. 

Endometrial carcinoma coexists with APAM in 8.8% of cases. APAM has an appearance similar to endometrial carcinoma, adenosarcoma, and carcinosarcoma, so its diagnosis is usually missed [5].

The origin for the majority of APAM cases (80%) is recognized at the cervical/isthmus level, but it can develop from any region of the uterus [6].

Research has demonstrated that the occurrence of uterine adenomyoma is associated with hormonal imbalances, environmental factors, previous cesarean sections, induced abortions, and other surgical procedures. These factors can lead to uterine enlargement and pelvic pain and may even result in infertility, thus significantly impacting the women’s reproductive health and overall quality of life [7].

While the specific relationship between atypical polypoid adenomyoma (APAM) and hormone therapy remains unclear, there have been reports of APAM occurring in patients who have received hormone therapies such as long-term tamoxifen therapy, hormonal substitution therapy, luteal hormone therapy, or clomiphene therapy. Therefore, it has been suggested that estrogen-related factors might have a significant role in the development of APAM [8].

## 2. Case Presentation

A 21-year-old patient, a virgin without a significant medical history, was addressed to our hospital for abundant vaginal bleeding. The local examination could not be performed due to her virginity. We performed abdominal and perineal ultrasounds, which showed a cervical tumor (Figure 1).

An MRI was performed that described a solid expansive formation, well delimited, with an inhomogeneous structure, with small cystic areas and an intense contrast socket. The mass was located in the vagina, apparently originating from the cervical level, with which it communicates over a distance of 4 mm, presenting the overall dimensions of 44/47/47 mm. The uterine body, the ovaries, and the pelvic organs were within normal limits. No images of pelvic adenopathies were revealed (Figure 2).

Surgery was recommended to the patient after the surgical defloration.

In the vagina, occupying the entire cavity, a mass was found, which bled on touch, presenting a pedicle originating above the cervix. We performed the excision of the mass using forceps, scissors, and a suture of the visible pedicle (Figure 3).

The close examination revealed a tumoral mass measuring 4.9/4.6/4.5 cm, with an irregular bosselated surface. The cut section of the tumor showed a white-grey surface with microcystic areas and a hemorrhagic gelatinous content.

The histopathological exam revealed the tissue fragments to be sufficiently delimited by a thin layer of the endometrium composed of a biphasic stromal and glandular proliferation and represented by cellular stroma with edematous areas with thin blood vessels interspersed with thick bundles of smooth muscular fibers that included irregularly contoured secretory endometrial glands (Figure 4 and Figure 5), organized in hyperplastic lobular areas, and lined by the simple and pseudostratified epithelium with hyperchromatic nuclei (Figure 6), eosinophilic metaplasia, and a tendency for squamous differentiation. 

Our microscopic findings are consistent with the following histopathological characteristics that define the atypical polypoid adenomyoma: a well-circumscribed biphasic tumor composed of endometrioid glands with a complex or lobular histoarchitecture, squamous morular metaplasia, and sometimes cytologic atypia, interspersed with a fibromyomatous stroma, which may present myxoid change. 

Four years post-surgery, the patient had no recurrences or abnormal vaginal bleeding. Moreover, she had no complementary medical treatment.

## 3. Discussion

### 3.1. Diagnostic: Imaging

At present, the gold standard for the diagnosis of uterine adenomyoma is a histopathological examination, but it has limitations due to the complicated operation, the long detection time, and the invasiveness involved, which makes it unsuitable for performing an early diagnosis of the disease. 

Various methods are available for clinically diagnosing adenomyosis, including salpingography, a magnetic resonance examination, and ultrasound examination. Salpingography, however, is susceptible to artifacts and may not ensure safety. The color Doppler ultrasound is a commonly used diagnostic tool in obstetrics and gynecology. It offers clearer and more precise imaging compared to the standard ultrasound and can rapidly and visually depict the two-dimensional distribution of blood flow. Additionally, it is characterized by good repeatability and noninvasiveness. The color Doppler ultrasound can be performed using the following two methods: transvaginal and transabdominal, providing flexibility in the diagnostic process [7].

The imaging and findings of polypoid adenomyoma are not extensively documented in the literature, as their characteristics can sometimes be mistaken for either prolapsed leiomyomas or malignancies. However, the presence of hemorrhagic cystic spaces within a prolapsed uterine tumor in the vaginal area should prompt consideration of the possible diagnosis of polypoid adenomyoma. The occurrence of such blood-filled cystic spaces is atypical and uncommon in leiomyomas and malignancies [9]. 

The ultrasound imaging of adenomyoma typically reveals several characteristic features, including variable degrees of uterine enlargement, the displacement of the endometrium, an indistinct boundary, the significant thickening of the uterine wall, and the presence of grid-like or strip-like hypoechoic areas of varying sizes. The blood flow signal is often not prominently visible, and there may be a cystic mass observed in the pelvic area. The transabdominal color Doppler ultrasound (TA-CDUS) is a conventional diagnostic method that provides a comprehensive view of the uterus and its adjacent organs. However, the transvaginal CDUS (TV-CDUS) has limitations, such as a relatively limited field of view, which may not capture the entire uterus, and it is less suitable for patients with concurrent vaginal bleeding and acute genital tract inflammation. 

The combined use of TA-CDUS and TV-CDUS proves effective in enhancing the visualization of the ultrasound signs associated with uterine adenomyoma. This approach enables a multidimensional and multi-angle examination of the uterus and its neighboring structures.

The color Doppler ultrasound demonstrates abundant blood flow within the tumor, but the peripheral blood supply may be insufficient. Patients with uterine adenomyoma tend to exhibit a higher resistance index (RI), systolic velocity (Vs), and pulsatility index (PI) values compared to those with uterine fibroids. This difference may be attributed to ectopic hyperplasia and the rich blood supply of endometrial tissues within the lesions.

A transvaginal ultrasound can indicate heterogeneous endometrial thickening, abnormal intrauterine echoes, and blood flow changes, although these findings lack specificity. Therefore, it is essential to differentiate atypical polypoid adenomyoma (APAM) from conditions such as endometrial polyps, endometrial cancer (EC), adenomyosis, uterine adenofibroma, and malignant mixed Mullerian tumors. In cases with clear indications, such as abnormal uterine bleeding, an abnormal intrauterine echo, and infertility, a hysteroscopy should be performed. The combination of transvaginal ultrasound and hysteroscopy plays a critical role in accurately identifying APAM lesions [10]. 

There are some reports of magnetic resonance image (MRI) findings of APAM [8]. 

When closely examined, polypoid adenomyomas do not exhibit any noticeable differences from ordinary endometrial polyps. However, on MRI imaging, they present well-defined intracavitary uterine masses and can extend into the lower uterine segment, endocervix, or uterine corpus. On T1-weighted images, they typically appear isointense. The signal intensity on the T2-weighted images may vary depending on the size and the presence of glands within the tumor.

A distinctive imaging feature that is particularly well-observed using an MRI is the identification of a visible stalk passing through the cervix, connecting a prolapsed mass back to the uterus. This is a valuable finding that suggests a prolapsed uterine tumor. Some of these tumors may also contain areas of hemorrhage within cystic spaces, which manifest as hyperintense foci on the T1-weighted images and are not suppressed on the fat-suppressed T1-weighted imaging. The enhancement pattern of polypoid adenomyomas is typically described as irregular or heterogeneous [9].

While an MRI and ultrasound are valuable tools for diagnosing polypoid adenomyomas, the definitive confirmation of this condition is obtained through the histopathological examination of the lesion [4].

There have been very few reported cases of positron emission tomography-computed tomography (PET-CT) findings in atypical polypoid adenomyoma (APAM). PET-CT images indicating an elevated uptake of fluorodeoxyglucose can raise suspicion for conditions like endometrial stromal sarcoma or atypical leiomyoma.

In such cases, it is crucial to implement careful follow-up with a combination of ultrasonography, hysteroscopy, and endometrial biopsy. However, the specific time intervals for these follow-up examinations have not been definitively established and may vary depending on the patient’s clinical context. Regular monitoring and diagnostic procedures are essential for the accurate evaluation and management of cases [8].

### 3.2. Immunohistochemistry and Molecular Diagnosis

Immunohistochemistry does not provide substantial assistance in distinguishing between atypical polypoid adenomyoma (APAM) and the myometrium infiltrated by carcinoma. This is because both the stromal component of APAM and the myometrium infiltrated by carcinoma exhibit positive reactions for desmin and smooth muscle actin [11]. Moreover, the stromal component of APAM presents immunoreactivity for SATB2 [12,13]. 

One potentially useful marker is CD10, as it tends to be absent in the stromal component of APAM, whereas the myoinvasive glands of endometrioid adenocarcinoma are typically surrounded by a thin rim of CD10-positive stromal cells, creating a ‘fringe-like’ staining pattern. H-caldesmon can also be informative, as it is strongly expressed in the myometrial smooth muscle invaded by the endometrioid adenocarcinoma, while the myomatous or fibromyomatous stroma of APAM typically lacks this expression [11].

The glands present a positive immunoexpression for beta-catenin, especially in squamous morules, which also express CDX2 and p16, as well as pancytokeratin, estrogen receptor, and the progesterone receptor [11,12,14]. 

Regarding the molecular aspects of atypical polypoid adenomyoma, there is limited information available. However, the identified abnormalities include MLH-1 promoter hypermethylation, PTEN and KRAS mutations, and microsatellite instability [14,15]. These findings are akin to the molecular abnormalities observed in the endometrial, including atypical hyperplasia and endometrioid adenocarcinoma. The presence of these molecular abnormalities, combined with the frequent association of APAM with atypical hyperplasia and/or endometrial carcinoma, suggests the possibility that APAM may represent a localized form of atypical hyperplasia.

In one study, the tumor markers CA125 and CA19.9 were examined, and they showed normal values; however, the evidence supporting their utility is limited. Instead, it is conceivable that certain markers associated with premalignant endometrial hyperplasia or endometrial cancer might be valuable in diagnosing and assessing the risk of APAM, given the shared molecular characteristics between these two conditions [3]. 

### 3.3. Histopathology

The diagnosis of atypical polypoid adenomyoma (APAM) is primarily based on a histological examination and is often challenging to distinguish macroscopically and clinically from conditions like endometrial polyps, submucous myomas, or adenofibromas.

Histologically, APAM displays a biphasic growth pattern characterized by sometimes atypical endometrial glands, with a squamous morular differentiation set against a background of abundant myofibromatous stroma. This histological pattern can resemble adenocarcinoma invading the myometrium or malignant mixed Mullerian tumors, adding to the diagnostic complexity of APAM [2,3].

Upon a low-power examination, atypical polypoid adenomyoma (APAM) exhibits endometrioid-type glands that can appear crowded or widely separated and are arranged haphazardly, often with a vague lobular architecture. These endometrioid glands may vary in appearance, being tubular or displaying complex branching. Squamous morules are not uncommon within the glands and may occasionally show central necrosis. The glands typically exhibit mild or, at most, moderate cytological atypia. In some instances, a ciliated or a mucinous epithelium may also be observed. These glands are embedded in an abundant myxoid or fibromyomatous stroma, which is often organized in short, interlacing fascicles and generally shows minimal cytological atypia, with only sporadic mitotic figures [2].

In cases with significant glandular crowding, distinguishing them from a grade I endometrioid adenocarcinoma can be challenging. The term “atypical polypoid adenomyoma of low malignant potential” has been proposed for lesions with a marked architectural complexity, but it is not widely adopted or recommended [2,3]. In such cases, if the tumor is not entirely removed, the risk of recurrence is higher. On the other hand, atypical polypoid adenomyoma with a low architectural complexity is typically considered benign when completely resected. Recurrence may still occur if there is an incomplete excision or when the architectural features closely resemble endometrial adenocarcinoma [2,3].

One of the primary challenges in the differential diagnosis, especially on a biopsy, is distinguishing APAM from an endometrioid adenocarcinoma that displays a myometrial invasion or is associated with a prominent desmoplastic stroma. These conditions can share some histological characteristics, making a precise diagnosis crucial for appropriate treatment and management [1].

### 3.4. Treatment

Given the rarity of APAM, no prospective trials have been performed, and the management of patients has never been standardized. As a result, there is no consensus on the optimal treatment and follow-up of APAM [3]. 

Opting for conservative treatment through operative hysteroscopy is often considered the best choice for managing atypical polypoid adenomyoma (APAM). This approach reduces the risk of recurrence, enhances the accuracy of diagnosing concurrent carcinoma or hyperplasia, and preserves the possibility of future pregnancies [6]. 

A hysteroscopic transcervical resection (TCR) is a common approach for preserving fertility in patients with atypical polypoid adenomyoma (APAM). However, a TCR carries the risk of uterine wall perforation, particularly when the tumor extends deeply into the uterine muscle layer. In cases where the tumor is extensive or deeply infiltrated, laparotomy tumor resection may be considered a fertility-sparing treatment option for APAM patients. Additionally, after surgery, the use of the LNG-IUS (levonorgestrel-releasing intrauterine system) may serve the following dual purpose: preventing adhesions and facilitating the disappearance of the tumor [16].

Atypical polypoid adenomyoma (APAM) tends to have high recurrence rates when treated conservatively. Therefore, hysterectomy has traditionally been considered the preferred treatment for this condition. However, given that APAM primarily affects premenopausal women in most cases, there is a need for fertility-sparing approaches. Several conservative treatment options have been adopted, including progestin-based hormonal therapy (HT) either with or without maintenance (M), a hysteroscopic transcervical resection (TCR), dilation and curettage (D and C), or a combination of HT with a TCR or D and C. These approaches aim to address APAM while preserving the patient’s reproductive potential [3]. 

In post-menopausal women, hysterectomy is often considered the preferred treatment for atypical polypoid adenomyoma (APAM). However, for patients who wish to retain their fertility potential, a more conservative surgical approach is typically recommended [17].

As the common age of APAM onset is approximately 40 years, fertility-sparing treatment is often required [16].

The fertility-sparing treatment options for atypical polypoid adenomyoma (APAM) include the prolonged use of high-dose progestogens and the trans-cervical resection of growth with the removal of an adequate portion of the uterine muscle layer, which can lead to positive fertility outcomes. In such cases, a long-term follow-up is essential, and regular hysteroscopic biopsies are recommended to monitor the patient’s condition [4]. 

While the precise cause of atypical polypoid adenomyoma (APAM) remains incompletely understood, it is generally believed that prolonged exposure to an estrogenic stimulation of endometrial stromal progenitor cells contributes to the development of APAM. As a result, a conservative treatment approach involving a local excision through a uterine curettage or a polypectomy, followed by progestational agents, has been proposed [4].

However, this conservative treatment method has some limitations, as persistent or recurrent lesions have been reported in 45% of patients who have undergone it [4]. It appears that a blunt curettage or a polypectomy may not be adequate for complete lesion resection, and the effectiveness of progestational agents remains uncertain.

Hysteroscopic surgery has been suggested as a more advantageous option in terms of successful pregnancy outcomes when compared to hormonal therapy alone. Hysteroscopic treatment eliminates the need for a uterine curettage, which is often required with hormonal therapy, and thus, the former approach can help avoid the potential damage to the endometrium [18]. 

The following four-step technique for managing atypical polypoid adenomyoma (APAM) has been proposed, involving a hysteroscopic resection of the lesion and a careful follow-up [3]: (1) the removal of APAM; (2) the removal of the endometrium surrounding the lesion; (3) the removal of the myometrium beneath the lesion; (4) random endometrial biopsies.

An alternative approach proposed by Nomura et al. (2018) involves medical treatment using medroxyprogesterone acetate in 18 patients, including 13 patients with APAM, 4 patients with APAM coexisting with endometrial adenocarcinoma, and 1 patient with APAM coexisting with atypical endometrial hyperplasia, all diagnosed after dilation and curettage [19]. Among the 18 patients studied, 14 achieved either a partial or complete response to this medical treatment, while 8 experienced recurrences. Ultimately, 10 of these patients underwent hysterectomy, and 9 were diagnosed with endometrial cancer. Notably, among patients under 35 years old, 4 out of 5 successfully had children [19].

Additionally, the use of a levonorgestrel intrauterine device (IUD) has been suggested as a form of maintenance therapy for young women who wish to preserve their fertility [17]. 

Medroxyprogesterone acetate (MPA) treatment for atypical polypoid adenomyoma (APAM) can be a reasonable option. However, it is important to note that recurrence may occur even after complete remission with the MPA treatment. The recurrence rate following a hysteroscopic transcervical resection (TCR) is lower than that after dilation and curettage (D and C). This difference can be attributed to the greater precision of a TCR in locating the lesion and reducing the likelihood of residual lesions. Nevertheless, even with the MPA therapy or the megestrol acetate therapy, recurrence rates of 10–33% have been reported [19].

One drawback of laparotomy tumor resection is the risk of uterine rupture during pregnancy. Consequently, the careful management of pregnancy and the consideration of an elective cesarean section may be necessary to mitigate this risk [16]. 

### 3.5. Follow-Up

A review that incorporated 46 observational studies and 296 cases in fertile women found that the prevalence of atypical polypoid adenomyoma (APAM) relapse was 44% [6]. Notably, the cases treated with operative hysteroscopy had a lower prevalence of relapse compared to the cases treated with blind curettage and polypectomy. Furthermore, the review reported that the prevalence of concomitant or follow-up diagnoses of endometrial carcinoma was 16% [6].

The risk of developing cancer during follow-up was significantly lower in the cases treated with hysteroscopy, with a new cumulative diagnosis rate of 10.56% at 5 years of follow-up, as opposed to the cases treated with blind curettage and polypectomy, which had a 35.5% new cumulative diagnosis rate at 5 years. Importantly, the review indicated that medical treatment with medroxyprogesterone acetate after surgery did not reduce the recurrence of APAM [6].

The review also reported that pregnancy was observed in 79% of cases in which the desire for pregnancy was expressed, highlighting the potential for fertility preservation in these cases [6]. 

Considering that atypical polypoid adenomyoma (APAM) has been linked to hormonal factors, it is essential to assess the effectiveness and safety of maintenance hormonal therapy following fertility-preserving treatment using medroxyprogesterone acetate (MPA).

The analysis suggested that maintenance hormonal therapy after MPA treatment is highly effective and safe, particularly for patients with a BMI (body mass index) of 24 kg/m^2^ or higher and those experiencing irregular menstrual cycles. This approach can help manage and prevent the recurrence of APAM while preserving fertility [19]. 

A hysteroscopic transcervical resection (TCR) has demonstrated a notably higher initial response rate compared to other conservative treatments for atypical polypoid adenomyoma (APAM). Both a TCR and a TCR combined with hormonal therapy (TCR + HT) resulted in higher final complete response rates and a lower progression rate when compared to hormonal therapy (HT) without maintenance [16].

Despite the advantages of a TCR, it is important to acknowledge that there is still a risk of recurrence (29.8%) and progression (10.8%), even after this procedure. In light of these risks, a hysterectomy might be considered the preferred treatment choice. However, this approach may not be suitable for patients who wish to preserve their fertility or have a desire for future pregnancies [3]. 

Without subsequent hormonal therapy, hysteroscopic transcervical resection (TCR) appears to be a more effective treatment than other conservative methods for atypical polypoid adenomyoma (APAM). In an overall assessment, a TCR alone is deemed an efficient and safe fertility-preserving option, with an initial response seen in 98.7% of treated patients. The outcomes for a TCR alone include a progression rate of 10.8%, a final complete response rate of 77.3%, a recurrence rate of 29.8%, and a pregnancy rate of 21.1% [3].

Compared to a TCR with HT, a TCR alone demonstrates significantly higher rates of initial response (98.7% versus 69.2%), with no significant differences in other outcome measures [3]. An advantage of a TCR without HT may be the shorter treatment duration, thus enabling earlier attempts at conception.

The four-step TCR technique is suggested as the optimal fertility-sparing treatment for women with APAM. To ensure the success of this approach, a long-term follow-up plan is advisable. This follow-up regimen typically includes dilation and curettage or a hysteroscopic biopsy combined with transvaginal ultrasonography every 3 months for the first 2 years, followed by monitoring every 4–6 months for an additional 3 years, and yearly thereafter [3].

## 4. Conclusions

Atypical polypoid adenomyoma (APAM) represents a very rare uterine lesion with a high recurrence rate and a risk for malignant transformation. The particularity of our case included the very young age of the patient, the absence of former engagement in sexual intercourse, the unusual size and localization of the mass, as well as the therapeutic approach of the tumor.

Considering the risk of recurrence and progression, atypical polypoid adenomyoma (APAM) may be managed through hysterectomy for patients who do not wish to preserve their fertility. A hysteroscopic transcervical resection (TCR) may be considered the first line of fertility-sparing treatment for APAM, given its superiority over other conservative approaches in terms of both effectiveness and safety. 

A comprehensive description of the ultrasound and MRI images, the introduction of a standardized classification system, and histopathological confirmation can simultaneously contribute to a better understanding of this disease’s impact and its influence on reproduction.

## Figures and Tables

**Figure 1 life-13-02352-f001:**
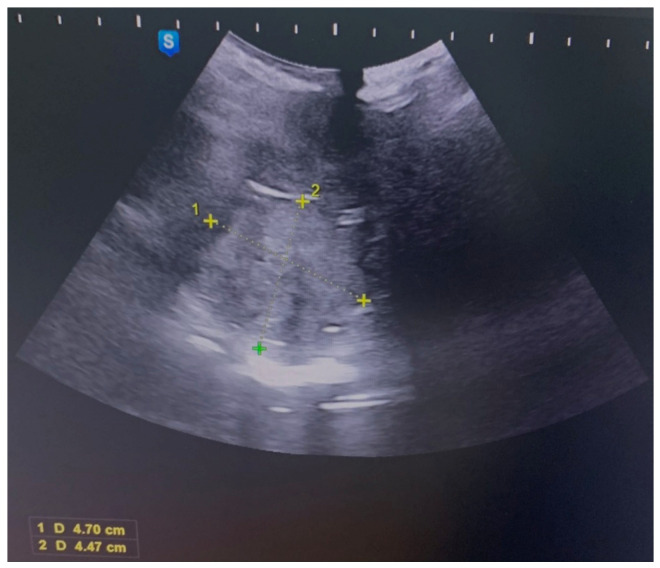
Abdominal ultrasound image of an irregular lesion occupying the vagina of 4.7 cm/4.47 cm.

**Figure 2 life-13-02352-f002:**
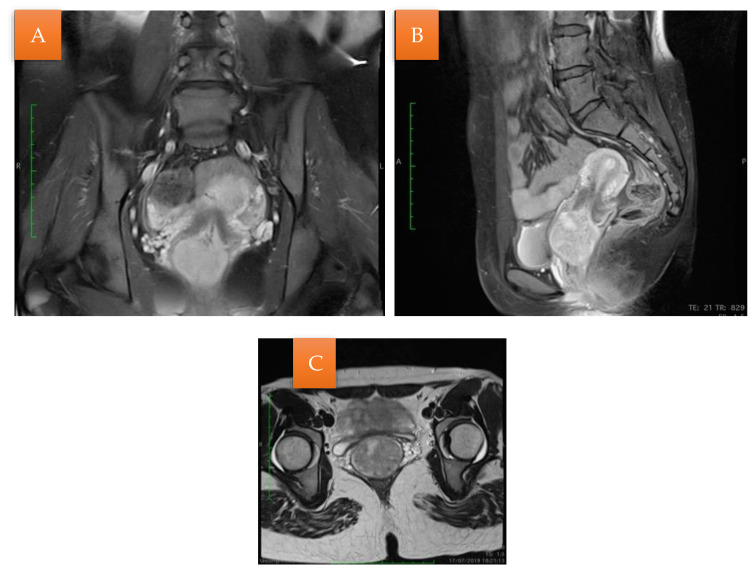
MRI examination (**A**); Sagittal section: the mass originating from the cervical level with small cystic areas (**B**); Coronal section: the vaginal mass with small cystic areas (**C**).

**Figure 3 life-13-02352-f003:**
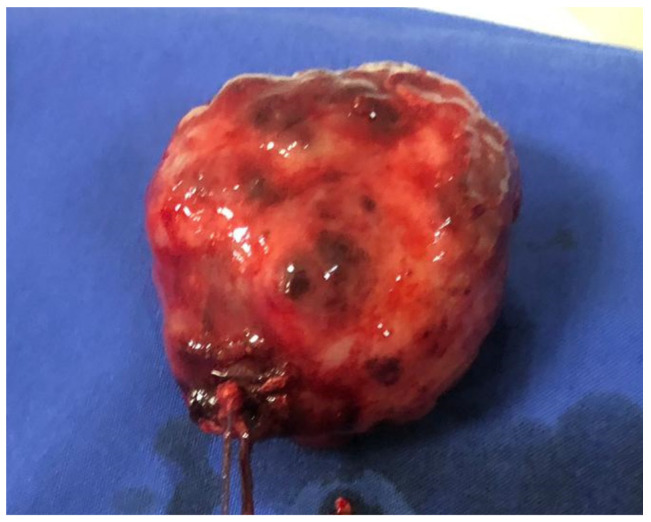
Bleeding tumoral mass presenting the overall dimensions of 4.9/4.6/4.5 cm, with an irregular/bosselated surface.

**Figure 4 life-13-02352-f004:**
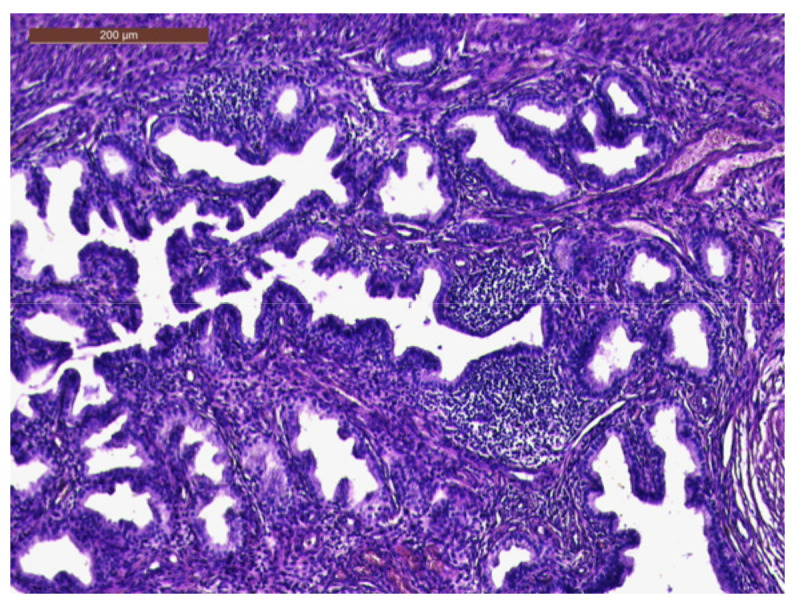
Endometrial glands with lymphoid follicles and associated smooth muscle fibers (HE × 10).

**Figure 5 life-13-02352-f005:**
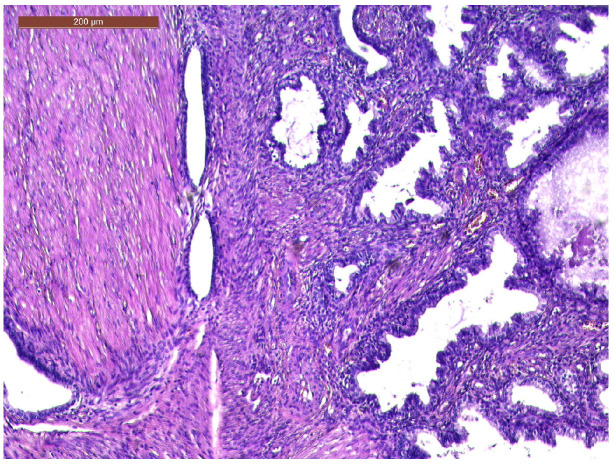
Thick bundles of smooth muscle fibers interspersed with endometrial areas containing irregularly oriented glands and dense cellular stroma (HE × 10).

**Figure 6 life-13-02352-f006:**
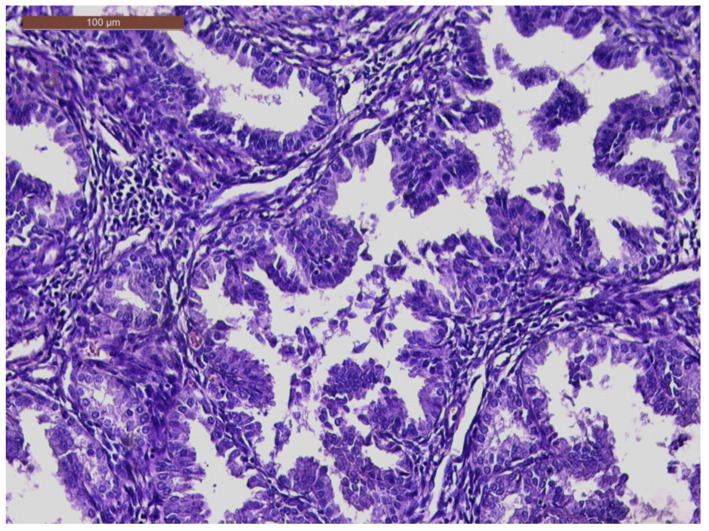
Hyperplastic endometrial glands, with irregular lumen; focal pseudostratified epithelium; and hyperchromatic nuclei (HE × 20).

## Data Availability

Data is contained within the article.

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
