# Peer review of "Prolapsed Atypical Polypoid Adenomyoma—A Case Report and Literature Review"

_life, 2023, doi:10.3390/life13122352_

Round 1
Reviewer 1 Report
Comments and Suggestions for Authors
See the atached file.

I suggest authors to carefully revise all throughout the text.
The manuscript must be reviewed and corrected by a native English speaker!
Author Response
Dear reviewer,
Firstly I want to thank you for reading and reviewing our work.
Please see the attachement

Reviewer 2 Report
Comments and Suggestions for Authors
Title is relevant for the content, but can be changed to give more scientific soundness and importance to the article. Introduction provides sufficient information, supported by up to date references, but the importance of this article and the therapeutic approach should be underlined in this chapter. Case report is present clear and supported by good images. Discussion offers an abundent literature review. Conclusions are supported by the data presented. Bibliography is up to date.
Comments on the Quality of English LanguageMinor english editing required
Author Response
Thank you very much for taking the time to read and review this manuscript. We have revised and corrected the whole text accordinglyReviewer 3 Report
Comments and Suggestions for Authors
The articles presents a rare pathology. Some observations
line 266:Additionally, after surgery, the use of the LNG-IUS (levonorgestrel- 265 releasing intrauterine system) may serve a dual purpose: preventing adhesions and 266 facilitating the disappearance of the tumor [10].
on the AAGL practice guidelines on preventing intrauterine adhesions is stipulated that if an IUD is used for adhesion prevention after ooperative hysteroscopy it should be inert not with progestins. This must be clarified
On discussion: this procedure was performed on a patient with desire to preserve her fertility. Anti adhesion options should be discussed after hysteroscoipic resection.
Comments on the Quality of English Languageno comments
Author Response
Thank you very much for taking the time to read and review this manuscript
Thank you for pointing this out. The use of LNG-IUS in these rare cases is only to reduce the risk of recurrences or malignant transformation and there are no standardized treatments till now. We did not perform any hysteroscopic resection in our case and we didn’t think of focusing our discussions on other methods of prevention
Reviewer 4 Report
Comments and Suggestions for Authors
The topic discussed in the publication concerns extremely rare tumors occurring in the uterus. The issue discussed in the text is an interesting medical problem worth publishing. However, the text requires some corrections:
1. The figures included in the text require better descriptions, highlighting the anatomical structures presented in the pictures.
2. The part of the text named: immunohistochemistry concerns not only immunohistochemical but also molecular diagnostics as well as tumor markers.
3. Presenting certain information (e.g. risk factors) in the form of a table will make the text more clear and reader-friendly.
4. The method of presenting the literature requires standardization in accordance with the requirements of the journal.
Comments on the Quality of English LanguageThe text should be corrected in terms of syntax and punctuation by a native English speaker.
Author Response
Thank you very much for taking the time to read and review this manuscript. Thank you for pointing this out. We agree with this comment and we have added the anatomical description We do agree that the subtitle did not cover all the discussed points. We have also changed the suggested subtitle to "immunohistochemical and molecular diagnosis". The whole manuscript was thoroughly checked and corrected accordingly in terms of syntax and punctuation
Reviewer 5 Report
Comments and Suggestions for Authors
What is the new aspect?
This is like a review article, not a case report.
Comments on the Quality of English LanguageModerate editing of the English language is required.
Author Response
Thank you very much for taking the time to read and review this manuscript.The particularity of our case consisted in the very young age of the patient, the absence of former engagement in sexual intercourse, the unusual size and localisation of the mass as well as the therapeutic approach of the tumor.
In the title we have have added that this is a case report followed by a literature review
Thank you for pointing that a revision of english is required. We have revised and corrected the whole text accordinglyRound 2
Reviewer 1 Report
Comments and Suggestions for Authors
See the attached file.

Reviewer 5 Report
Comments and Suggestions for Authors
The authors' resubmitted paper has been sufficiently revised for publication in the journal. I believe that "accept" is appropriate in the present form.